# Ligamentization of the reconstructed ACL differs between the intraarticular and intraosseous regions: A quantitative assessment using UTE-T2* mapping

Rikuto Yoshimizu[1], Junsuke Nakase[1]*, Miho Okuda[2], Kazuki Asai[1], Mitsuhiro Kimura[1], Tomoyuki Kanayama[1], Yusuke Yanatori[1], Hiroyuki Tsuchiya[1]

1 Department of Orthopedic Surgery, Graduate School of Medical Sciences, Kanazawa University, Kanazawa, Japan, 2 Department of Radiology, Kanazawa University Hospital, Kanazawa, Japan

* nakase1007@yahoo.co.jp

**Data Availability Statement:** All relevant data are within the paper and its Supporting Information files.

## Abstract

### Background

The purpose of this study was to prospectively observe the trends of ultrashort echo time (UTE)-T2* values for the intraarticular and intraosseous regions of reconstructed anterior cruciate ligaments from 6 to 12 months after anterior cruciate ligament reconstruction by using UTE-T2* mapping, and to investigate the changes and differences over time in each region.

### Methods

Ten patients underwent UTE-T2* mapping of the operated knee at 6, 9, and 12 months after anterior cruciate ligament reconstruction. The UTE-T2* values of intraarticular and intraosseous regions of reconstructed anterior cruciate ligaments at 6, 9, and 12 months postoperatively were statistically compared.

### Results

The UTE-T2* values of the intraarticular region at 6 months postoperatively were significantly higher than those at 9 and 12 months. There were no significant differences in the UTE-T2* values at 6, 9, and 12 months postoperatively in the intraosseous region. At 6 months postoperatively, the UTE-T2* values of the intraarticular region were significantly higher than those of the intraosseous region. The UTE-T2* values of the intraosseous region at the tibia were significantly lower than those of the other sites at any postoperative time point.

### Conclusions

According to UTE-T2*mapping-based findings, histological maturation of reconstructed ACLs is faster in the intraosseous region than in the intraarticular region. In particular, the intraarticular region is still undergoing rapid histologic changes at 6 months postoperatively,

**Funding:** The authors received no specific funding for this work.

**Competing interests:** The authors have declared that no competing interests exist.

and its tissue structure is less substantial than normal. The findings of this study may provide clues to determine the optimal timing for safe return to sports in terms of ligamentaization of reconstructed ACLs.

## Introduction

Patients who undergo primary anterior cruciate ligament (ACL) reconstruction (ACLR) are at a high risk of ipsilateral retear in the first 12 months, and early return to sport (RTS) after ACLR is one of the most important risk factors [1–4]. The ligamentization status of the reconstructed ACL is an important factor when considering RTS [3–5]. Reconstructed ACLs undergo cytological rearrangement and adapt to their biological and mechanical environment over time after ACLR [6]. These findings are based on samples obtained from animals and humans, but not all results from studies with animals can be correlated with physiological changes in humans [6–8]. Histological monitoring of the reconstructed ACL may be ideal, but ethical considerations limit the possibility of a second look and tissue harvesting in patients showing good progress, and the evaluations are limited to the collection site [3,4]. Magnetic resonance imaging (MRI) has been used to evaluate the ligamentization process of the reconstructed ACL to compensate for these limitations.

MRI signals, such as the signal-to-noise quotient (SNQ) and median signal intensity (SI), reflect the biological processes underlying cell proliferation and extracellular matrix remodeling, and have been considered useful for assessing the ligamentization process of reconstructed ACL [5,9–11]. However, these MRI signals show problems related to accuracy and quantification. A previous review has shown that the SNQ and SI of reconstructed ACLs vary significantly even in studies with similar imaging acquisition protocols and postoperative time points [9]. This can be attributed to the fact that the conventional MRI signal intensity is affected by the image sequence and scanner hardware. Moreover, Tendon and ligaments normally have short T2 relaxation times leading to low MRI signal in conventional MRI protocols [3–5,9].

Relaxation time T2$^*$ reflects the intrinsic property of tissue and should be independent on image sequence and acquisition parameters [12]. These variables reflects the T2$^*$ relaxation of bounded water with collagen of tendons and ligaments, and an ideal to capture the changes in tissue structure and organization during ligamentization of reconstructed ACL [3,4,13]. The ultrashort echo time (UTE) pulse sequence is a method of acquiring data immediately after excitation by using short radiofrequency pulses. By acquiring multiple echoes, the UTE-T2$^*$ relaxation time of tendons and ligaments, which cannot be assessed in conventional gradient-echo based MRI assessment due to sub-ms T2$^*$ values of collagen-bound water [13–15]. Thus, the UTE T2$^*$ technique is an excellent tool for observing the ligamentization process of a reconstructed ACL [3,4].

The critical time for RTS after ACLR is 6–12 months postoperatively, but data evaluating the ligamentization process of the reconstructed ACL by using the UTE-T2$^*$ technique are extremely limited [3,4]. Although the intraarticular and intraosseous regions of reconstructed ACLs undergo different maturation processes, the difficulties in biopsy have limited the available knowledge of the intraosseous region of reconstructed ACLs. Although the influence of the differences in each process in the images is important, no study has evaluated the maturation process of the intraosseous region of reconstructed ACLs by using the UTE-T2$^*$ technique.

To address these aspects, the purpose of this study was to prospectively observe the trends in UTE T2$^*$ values for the intraarticular and intraosseous regions of reconstructed ACLs from

6 to 12 months after ACLR by using UTE-T2* mapping, and to investigate the changes and differences over time in each region. We hypothesized that the UTE T2* values and their trends would differ for the intraarticular and intraosseous regions of reconstructed ACLs at each postoperative time point.

## Patients and methods

### Patient selection

The study design was approved by the Ethical Committee of the Graduate School of Medical Sciences, Kanazawa University (#2936), and conducted in accordance with the principles expressed in the Declaration of Helsinki. Patients who underwent initial ACLR with hamstring tendons during 2018–2020 were eligible for inclusion. The purpose of this study was explained to the participants, and written informed consent was obtained from the participants or their parents. Patients with a history of ipsilateral or contralateral knee injury or surgery and those who were unable to attend the hospital or undergo MRI were excluded.

Ten female patients were enrolled, and they underwent UTE-T2*mapping of the operated knee at 6, 9, and 12 months postoperatively. The mean age of participants at the start of the study was 18.4 ± 4.3 years. The mean body mass index (BMI) was 21.7±2.1 kg/m$^2$ and the mean time to surgery was 42.1 ± 11.3 days. Five patients had right knee injuries and five had left knee injuries, and all were non-contact injuries.

**Surgical procedure.**   All patients were treated by a single orthopedic surgeon specializing in arthroscopy. In all cases, the transplanted tendon was created with a single bundle of semitendinosus or semitendinosus and gracilis tendons. The femoral tunnel was created in the middle of the anatomical ACL footprint by the inside-out method using a rounded rectangular dilator [16]. The tunnel on the tibial side was created in a circular shape in the middle of the anatomical ACL footprint. The femoral side of the graft tendon was fixed using a cortical device (Tight Rope; Arthrex, USA). After the graft was pretensioned several times, the tibial side was fixed using a tibial fixation implant (Tension-Loc; Arthrex, USA). All patients underwent rehabilitation using a standardized postoperative protocol.

**Imaging procedure.**   A clinical 1.5-T MRI scanner (Ingenia 1.5 T CX; Philips Healthcare, Best, The Netherlands) and an eight-channel receiver knee coil were used for all patients. The UTE-T2*maps were calculated via monoexponential fitting of a series of T2*-weighted MR images, which were acquired using the 3D fast-field echo technique. The typical acquisition parameters were as follows: slice thickness, 3 mm; number of slices = 45; field of view = 16 cm; echo time (TE)/repetition time (TR) = 0.14, 4.74, 9.34, and 13.94 ms/29 ms; flip angle = 25˚; acquisition matrix = 272 × 272; bandwidth = 522 Hz; and scan time = 9 min 31 s. Four sets of images were obtained using a single four-echo UTE acquisition. Images were obtained from a picture archiving and communication system. T2* maps were directly calculated on a pixel-by-pixel basis by using a monoexponential fitting algorithm available on the scanner. The equation is expressed as follows: SI(TE) = S0*exp(-TE/T2*), where SI(TE) is the single intensity at each TE, and S0 is the equilibrium magnetization.

The slices in the UTE-T2* map for measuring T2* values were selected by referring to the slice in which the reconstructed ACL was more distinct in the oblique sagittal T2-weighted image. The UTE-T2* values for the intraarticular region of the reconstructed ACLs were measured at three sites based on the method previously reported by Okuda et al [15]: proximal, middle, and distal. Values for the intraosseous regions of the reconstructed ACLs were measured at one site each in the tibia and femur. One orthopedic surgeon (RY, Observer 1) used a 5–10 mm$^2$ circle to manually segment the regions of interest (ROIs) within areas unaffected by artifacts (Fig 1). All measurements were taken three times, and the average value was used as

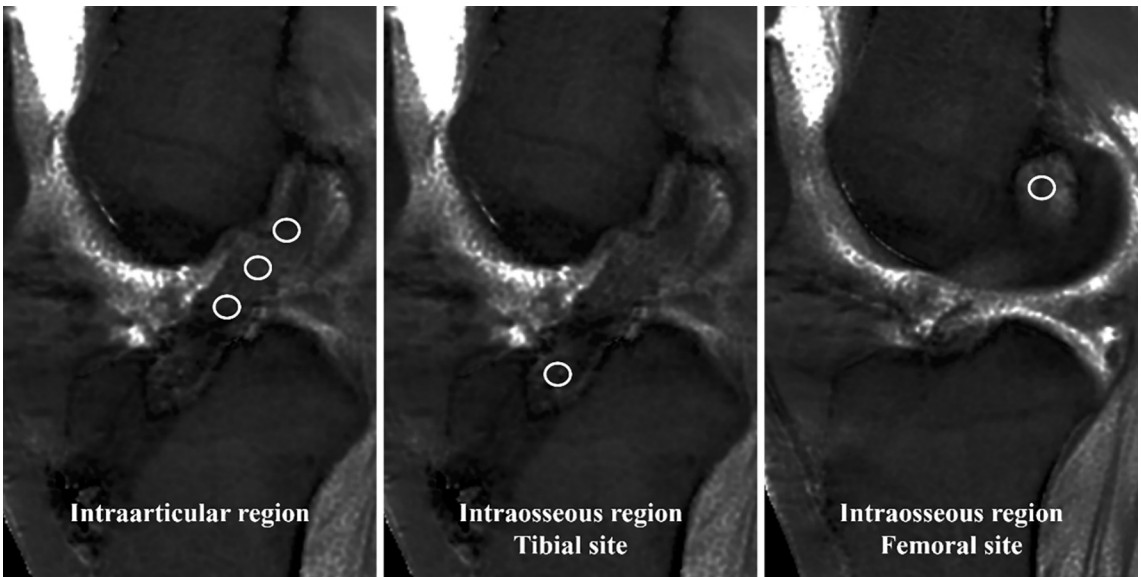

**Fig 1. Measurement of the UTE T2\* values.** The UTE-T2\* values for the intraarticular region of reconstructed ACLs were measured at three sites, and those for the intraosseous region of reconstructed ACLs were measured at one site each. The regions of interest (ROIs) for each site were segmented at the areas unaffected by artifacts by using a 5–10 mm$^2$ circle.

the UTE-T2\* value for each region. The UTE-T2\* values of the intraarticular region were calculated by further averaging those of the proximal, middle, and distal sites. To assess interobserver reliability, another orthopedic surgeon (YY, Observer 2) independently performed measurements using the same method.

**Statistical analysis.** All statistical analyses were performed using IBM SPSS Statistics for Windows, version 27.0. The UTE-T2\* values for the intraarticular and intraosseous regions of reconstructed ACLs at 6, 9, and 12 months postoperatively were compared using one-way analysis of variance (ANOVA). The UTE-T2\* values for the intraarticular and intraosseous regions of reconstructed ACLs at each postoperative month were also compared using ANOVA. Statistical significance was set at P < 0.05. The intra- and interobserver reliabilities (intraclass correlation coefficient [ICC]) of the UTE-T2\* values at 6 months after ACLR were calculated, and the measured values were rated as follows; 0.00–0.40, poor; 0.41–0.75, fair to good; and 0.76–1.00, good to excellent. Sample size was calculated using G-power 3.1 (effect size, 1.3; α-error, 0.05; and target power, 0.8); a minimum of nine participants was recommended on the basis of a previous study [4].

## Results

The UTE-T2\* values of the intraarticular region of reconstructed ACLs were 13.1 ± 1.9 ms, 11.7 ± 1.5 ms, and 11.1 ± 1.3 ms, respectively, at 6, 9, and 12 months postoperatively, and the UTE-T2\* value at 6 months postoperatively was significantly higher than those at 9 and 12 months (P < 0.01 vs. 9 months; P < 0.01 vs. 12 months). Compared to the UTE-T2\* value of the normal ACL reported previously, the T2\* value at 6 months postoperatively was significantly higher (P<0.01) (Fig 2). In the intraosseous region of reconstructed ACLs, the UTE-T2\* values at the tibial site were 7.4 ± 1.2 ms, 7.1 ± 1.0 ms, and 6.9 ± 1.2 ms, respectively, at 6, 9, and 12 months postoperatively, with no significant difference between the values at 6 and 9 months (P = 0.44), 6 and 12 months (P = 0.20), and 9 and 12 months (P = 0.85). The UTE-T2\* values at the femoral site were 11.5 ± 2.4 ms, 11.0 ± 1.7 ms, and 11.1 ± 1.5 ms at 6, 9,

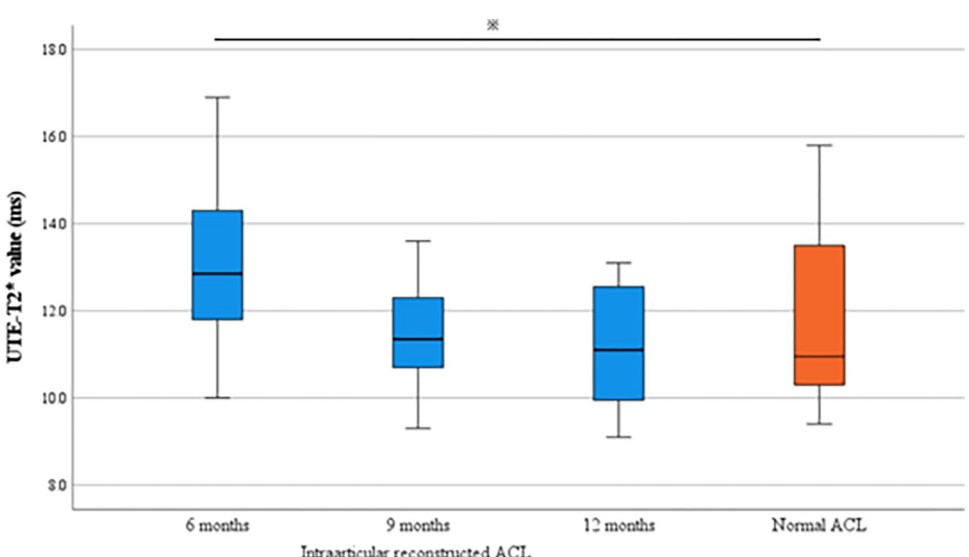

**Fig 2. Boxplot showing the UTE-T2* values for the intraarticular region of reconstructed ACLs between 6 to 12 months after ACLR, relative to the values for the normal ACL.** The UTE-T2 values for the intraarticular region were comparable to those for the normal ACL from 9 months postoperatively. UTE, ultrashort echo time; ACL, anterior cruciate ligament; ACLR, anterior cruciate ligament reconstruction.

and 12 months postoperatively, also showing no significant difference between the values 6 and 9 months (P = 0.56), 6 and 12 months (P = 0.72), and 9 and 12 months (P = 0.97). At 6 months postoperatively, the UTE-T2* values were significantly higher for the intraarticular region of reconstructed ACLs than for the intraosseous region of reconstructed ACLs (P < 0.01 vs. tibial site; P < 0.01 vs. femoral site). The UTE-T2* values at the tibial site for the intraosseous region were significantly lower than those at the other sites at all time points (9 months: P < 0.01 vs. femoral site, P < 0.01 vs. intraarticular; 12 months: P < 0.01 vs. femoral site, P < 0.01 vs. intraarticular) (Table 1, Fig 3). Interobserver reliability was good to excellent for both intraarticular and intraosseous regions in the segmentation and registration process at 6 months after ACLR (ICC, 0.84; 95% CI, 0.57–0.93; tibial site: ICC, 0.94; 95% CI, 0.88–0.97; femoral site: ICC, 0.76; 95% CI, 0.50–0.89). Intraobserver reliability at 6 months after ACLR was also good to excellent in all regions (intraarticular: ICC, 0.97; 95% CI, 0.93–0.99; tibial site: ICC, 0.85; 95% CI, 0.65–0.96; femoral site: ICC, 0.93; 95% CI, 0.83–0.98).

**Table 1. UTE-T2* values (ms) for each region at 6–12 months after ACLR.**

|  | 6 months | 9 months | 12 months |
|---|---|---|---|
| Intraarticular region of reconstructed ACL |  |  |  |
| Distal | 12.3±1.4 | 10.6±1.6 | 10.3±1.5 |
| Middle | 13.1±2.2 | 12.1±1.7 | 11.4±1.7 |
| Proximal | 13.8±2.6 | 12.6±1.8 | 11.7±1.5 |
| Mean | 13.1±1.8 | 11.7±1.5* | 11.1±1.3* |
| Intraosseous region of reconstructed ACL |  |  |  |
| Tibia site | 7.4±1.2 | 7.1±1.0 | 6.9±1.2 |
| Femoral site | 11.5±2.3 | 11.0±1.6 | 11.1±1.5 |

Values are presented as mean ± SD. UTE, ultrashort echo time; ACL, anterior cruciate ligament.

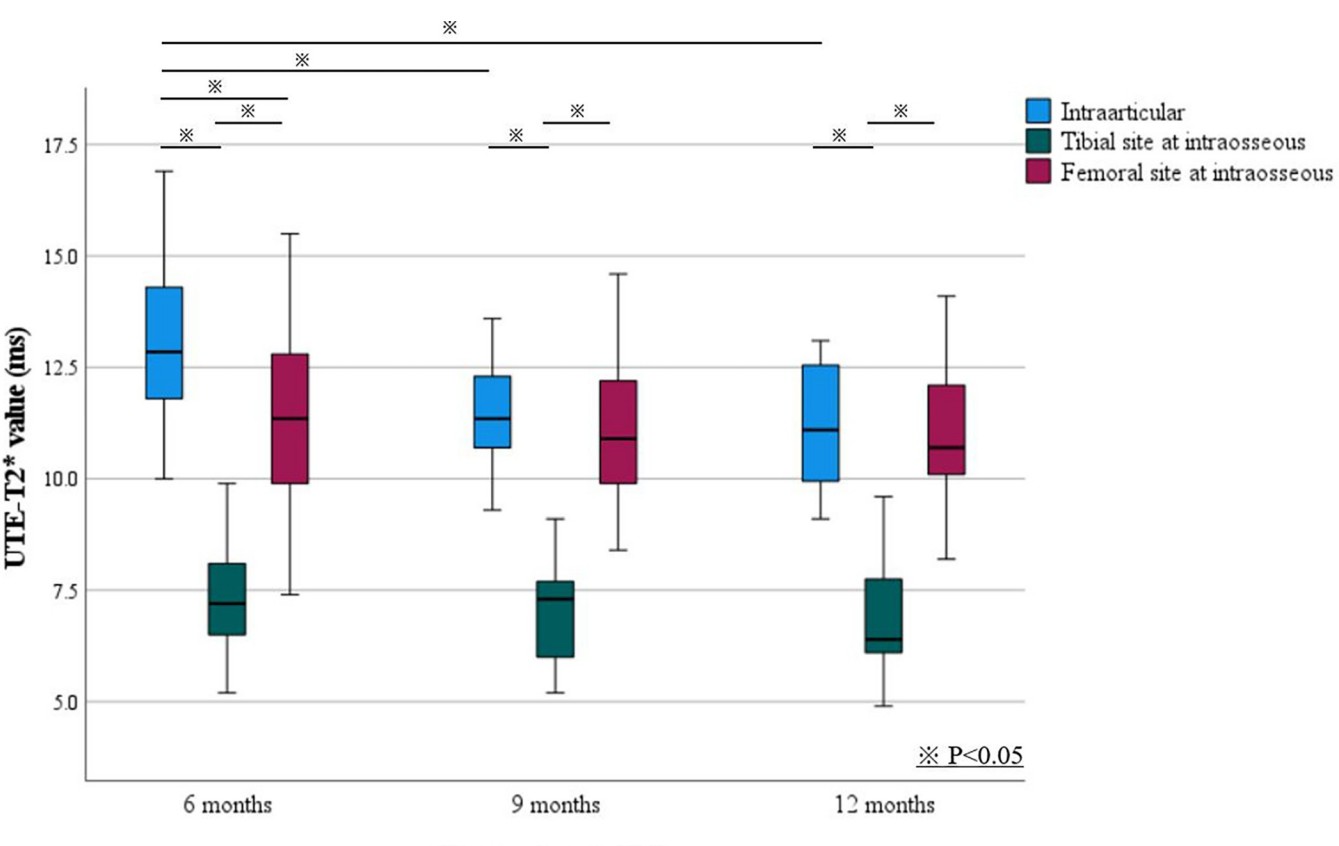

**Fig 3. Boxplot of changes in UTE-T2* values for the intraarticular and intraosseous regions of reconstructed ACLs between 6 to 12 months after ACLR.** The UTE-T2* values for the intraarticular and intraosseous regions show different transitions. UTE, ultrashort echo time; ACL, anterior cruciate ligament; ACLR, anterior cruciate ligament reconstruction.

## Discussion

In this study, we investigated the changes in T2* values for the reconstructed ACLs at 6, 9, and 12 months after ACLR by using UTE-T2* mapping separately for the intraarticular and intraosseous regions. The most important finding of this study was that the intraarticular region of reconstructed ACLs showed significantly lower UTE-T2* values from 6 to 9 months postoperatively, while the values for the intraosseous region of reconstructed ACLs did not change significantly. The UTE-T2* values at the tibial site in the intraosseous region of reconstructed ACLs were significantly lower than those at the femoral sites and the intraarticular regions at all time points.

Previous studies using UTE-T2* techniques in patients after ACLR have focused on assessment of changes in the knee cartilage and meniscus over time [17–19]. UTE-T2* mapping is suitable for assessing the ligamentization process of the reconstructed ACL because it can image organized collagen structures and capture microscopic changes [3,4,13–15]. However, there are limited data regarding UTE-T2* assessments of the reconstructed ACL, and the results of this study may provide an insight into the ligamentization process of the reconstructed ACL.

The signal intensities acquired from long-TE sequences used in conventional MRI vary depending on the acquisition protocol, software, and inherent parameters [3–5,9]. In contrast,

UTE-T2*mapping is an innovative tool for evaluating the ligamentization process of the reconstructed ACL because it shows no such limitations and can provide objective data [3,4].

Previous UTE-T2* studies have reported that UTE-T2* values for the intraarticular region of reconstructed ACLs increase rapidly until about 6 months and then slowly decrease, which is consistent with the findings of conventional MRI studies [3,4,9–11]. In the early postoperative period, the reconstructed ACLs undergo a stepwise and combined process of increasing fibroblast number, intense revascularization, and disintegration of collagen fibrils and their orientation. In particular, it has been shown that the regular collagen orientation and crimp pattern of reconstructed ACLs are lost in the early postoperative period and slowly restored only during the remodeling phase [6]. In other words, the increase in UTE-T2* values up to 6 months postoperatively may reflect the disruption of the collagen matrix in the reconstructed ACL in the early postoperative period, and the subsequent changes in the UTE-T2* values may be due to the transition to the main phase of remodeling [3,4,6–8]. The key question is how long the significant histological changes in the reconstructed ACL, that is, the rapid changes in UTE-T2* values, will continue. In this study, the UTE-T2* values for the intraarticular region of reconstructed ACLs decreased significantly from 6 to 9 months after surgery, during which time the remodeling changes in the reconstructed ACLs may occur rapidly. Warth et al. showed that the UTE-T2* values decreased significantly from 6 to 9 months in 10 patients after ACLR with hamstring or patellar tendons [4]. This is the only study that observed changes in the UTE-T2* values for reconstructed ACLs at 6, 9, and 12 months postoperatively, which supports the findings of the previous study. Chu et al. observed the evolution of UTE-T2* values for the intraarticular region of reconstructed ACLs and reported a significant decrease from 1 year to 2 years postoperatively, but the change was clearly slower than from 6 months to 1 year postoperatively [3]. Histologically, the reconstructed ACL undergoes structural changes up to 2 years postoperatively, and the structural changes stop at a stage of microstructure that is strictly different from that of the normal ACL [20]. In summary, the reconstructed ACL at 6 to 9 months postoperatively continues to undergo rapid tissue changes during the remodeling phase, and the tissue structure is unstable. After that time, the speed of histological changes declines, and the tissue structure becomes relatively stable.

As mentioned earlier, we had previously investigated the UTE-T2* values of the normal ACL in 12 healthy knees by using the same measurement methods used in this study [15]. When the results of this study were compared with those obtained for a normal ACL by ANOVA, the UTE-T2* value of the normal ACL was 11.9 ± 2.4 ms, which differed significantly from the value obtained 6 months postoperatively in the intraarticular region of the reconstructed ACL but not from the values obtained at 9 and 12 months postoperatively (Fig 3). This result suggests that the tissue structure of the reconstructed ACL is less substantial than that of the normal ACL at 6 months postoperatively. Previous studies comparing the UTE-T2* values for the intact ACL of the contralateral knee and reconstructed ACL showed no difference in the UTE-T2* values at 6 months postoperatively, but they did not consider influences of ACL injury and ACLR on the intact ACL of the contralateral knee [4]. The results of this study suggest that the histological structure of a reconstructed ACL is different from that of a normal ACL, but it is possible to reach a similar histological structure 9 months postoperatively.

The maturation processes of the intraarticular and intraosseous regions of reconstructed ACLs differ in relation to the biological processes at the early stages after ACLR [6–8,21,22]. Specifically, the intraarticular region of a reconstructed ACL undergoes revascularization from synovial fluid, and the intraosseous region of the reconstructed ACL undergoes revascularization from the adjacent cancellous bone. In a recent review, these processes were completed by 3 to 6 months postoperatively, with no significant difference in the rate of progression [6–8,21,22]. In this study, the UTE-T2* values of the intraosseous region of reconstructed ACLs

remained unchanged from 6 to 12 months postoperatively and were significantly lower than those for the intraarticular region of reconstructed ACLs, especially at 6 months postoperatively. This may indicate that histological maturation of the reconstructed ACL is faster in the intraosseous region than in the intraarticular region. In the intraosseous region, a fibrous interface in continuous contact with the bone tunnel was formed and stress-shielded by 6 months postoperatively, and this stress-shielding may have provided an advantage in histological maturation [21,22].

Furthermore, the UTE-T2* values were significantly lower at the tibial site in the intraosseous region of reconstructed ACLs than at the femoral site at all time points. Ahn et al. investigated the maturation in the intraosseous region of reconstructed ACLs using conventional MRI with SNQ, which also showed significant maturation of the tibial site [11]. Microvessels derived from the fat pad and posterior synovial tissue were considered to form a rich vascular envelope that contributes to the numerous intra-ligamentous branches for perfusion at the graft site. This may also reflect the mechanical environment at each site. Rodeo et al. investigated the relationship between tunnel motion in the reconstructed ACL and histological maturation of the femoral and tibial sites [23]. They concluded that the histological maturation of the reconstructed ACL in the femoral tunnel is inversely proportional to the magnitude of the graft tunnel motion, which affects local histological maturation.

This study had some limitations. First, although we limited the measurement of UTE-T2* values to the part where the influence of artifacts was suppressed, the influence of joint edema and magic angle effects was unavoidable. In particular, in the reconstructed ACL on the femoral side, the magic angle effect may be a factor responsible for overestimation of the UTE-T2* values due to the large bending angle. Second, we used only mono exponential UTE-T2* mapping techniques. Using mono exponential T2* fitting, the estimated apparent T2* values as reported in this study may include the effect from changes of bound/unbound tissue water fraction during ACL ligamentization. The fraction of bound water is quite high in both reconstructed and normal ACLs and include a first echo (TE = 0.1 ms), which may introduce errors in estimation of the mono-exponential map. In this study, bi-exponential UTE-T2* mapping was deemed inappropriate for routine clinical MRI because of its long acquisition time, although partial acquisition could have reduced the time [24,25]. Third, factors affecting the UTE-T2* values of reconstructed ACLs and their correlations with clinical and functional outcomes were not evaluated. We could not control for potential factors that may have influenced the results of this study. Although the participants enrolled in this study were women who underwent single-bundle ACLR with hamstring tendons, sex and the surgical technique including graft selection, may have influenced the results of this study. Future studies involving larger sample sizes and incorporating covariates that may affect the maturity of reconstructed ACLs are warranted. Investigation of the relationship of UTE-T2* values with clinical and functional outcomes is also important to provide ideas on safe return to sports in terms of maturation of reconstructed ACLs.

Despite these limitations, this is one of the few studies to investigate the graft maturation process in both intraarticular and intraosseous regions of reconstructed ACLs using UTE-T2* mapping. The objective data obtained in this study will play an important role in understanding the overview of the ligamentization process in a reconstructed ACL.

## Conclusions

According to UTE-T2* mapping-based findings, histologic maturation of reconstructed ACLs is faster in the intraosseous region than in the intraarticular region. In particular, the intraarticular region of ACLs is still undergoing rapid histological changes at 6 months

postoperatively, and its tissue structure is less substantial than normal. The findings of this study may provide clues to determine the optimal timing for safe return to sports in terms of ligamentaization of reconstructed ACLs.

## Supporting information

**S1 Data. Measurement data.**
(XLSX)

## Acknowledgments

This study would not have been possible without the cooperation of our collaborators. We would like to express our gratitude to the staff who helped us capture MRI images.

## Author Contributions

**Data curation:** Rikuto Yoshimizu, Kazuki Asai, Mitsuhiro Kimura, Tomoyuki Kanayama, Yusuke Yanatori.

**Formal analysis:** Rikuto Yoshimizu, Yusuke Yanatori.

**Investigation:** Rikuto Yoshimizu.

**Methodology:** Rikuto Yoshimizu, Miho Okuda, Hiroyuki Tsuchiya.

**Project administration:** Rikuto Yoshimizu, Junsuke Nakase, Miho Okuda, Hiroyuki Tsuchiya.

**Supervision:** Junsuke Nakase, Miho Okuda, Hiroyuki Tsuchiya.

**Writing – original draft:** Rikuto Yoshimizu.

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
