## [Decision Letter · Decision Letter 0]

25 Apr 2022

PONE-D-22-05680Ligamentization of the reconstructed ACL differs between the intraarticular and intraosseous regions: A quantitative assessment using UTE-T2* mappingPLOS ONE

Dear Dr. Nakase,

Thank you for submitting your manuscript to PLOS ONE. After careful consideration, we feel that it has merit but does not fully meet PLOS ONE’s publication criteria as it currently stands. Therefore, we invite you to submit a revised version of the manuscript that addresses the points raised during the review process.

We look forward to receiving your revised manuscript.

Kind regards,

Markus Geßlein, Ph.D.

Section Editor

PLOS ONE

Journal Requirements:

Additional Editor Comments:

Dear Authors,

thank you very much for submitting your well-performed study to PlosOne. After review by two experienced colleagues in this field I would kindly ask your for some improvments in your manuscript. You will find the detailled reviewers comments below.

Furthermore, I would kindly ask you to adress the additional issues below. I think this will make your manuscript more interesting for a broader readership.

Abstract

Conclusion

- Is a bit of repetition according to the abstracts result section? Please provide any potential clinical benefit of your study or describe why this finding could be important for future research (see comment on discussion below).

- Conclusion in the abstract and the manuscript should be same.

Material & Methods

-Please provide more information about the female patients included e.g. BMI, Height, side of injury, time from injury to surgery, maybe injury mechanism etc..

Discussion

-The first paragraph of the discussion should contain the most important findings of your study. I would suggest using the actual conclusion as first paragraph for the discussion. Then rewriting the conclusion with reference to the clinical importance or future research topics you would suggest.

-I would like to make the manuscript more interesting to read for orthopedic surgeons having to deal with the RTA/RTS issues in daily routine.

-I would like to ask you to move figure 3 to the results section. Further, please explain in the methods section how “normal ACL” values were aquired.

Limitations

-You did not provide data about the stability of knee after surgery. Orthopedic Surgeons well know that technical errors during surgery may result in decreased stability and that might affect the biological graft transformation. If you could provide data about the knee stability (clinical like Lachmann or KT 1000) that would improve the findings. If you don not have that data please mention this fact in the limitations.

-Please add information how the choice of enrolled participants could have influenced the results (only females..time between injury and surgery etc..)

-Did metal remnants from tunnel drilling interfere with the scan /measures?

Reviewers' comments:

Reviewer's Responses to Questions

**Comments to the Author**

1. Is the manuscript technically sound, and do the data support the conclusions?

Reviewer #1: Yes

Reviewer #2: Yes

2. Has the statistical analysis been performed appropriately and rigorously? 

Reviewer #1: Yes

Reviewer #2: Yes

3. Have the authors made all data underlying the findings in their manuscript fully available?

Reviewer #1: Yes

Reviewer #2: Yes

4. Is the manuscript presented in an intelligible fashion and written in standard English?

Reviewer #1: Yes

Reviewer #2: Yes

5. Review Comments to the Author

Reviewer #1: In this manuscript, authors investigated the maturation of reconstructed ACL using UTE-T2* imaging. This is a well-written manuscript. I do have some minor concerns:

1. Statistical Analysis: Only pairwise comparison of T2* values was conducted at three time points. As demonstrated in Figure 2, there are some clear trends of T2* over time, especially in intraarticular region. Correlation T2* values with recovery time would be interesting.

2. In Discussion, authors claimed that this is the first study to investigate regional graft maturation with UTE. There is at least one recent publication in this topic: Fukuda T, et al, Abbreviated quantitative UTE imaging in anterior cruciate ligament reconstruction, BMC Musculoskeletal Disords. 2019;20(1):426;

3. In Discussion, authors claimed the long acquisition duration needed for bi-exponential UTE study. It would be true if using fully-sampled acquisition. As described in Fukuda et al paper, it is possible to reduce the time by partial acquisition.

4. The confounds of mono-exponential instead of bi-exponential should be discussed. The fraction of bound water is quite high in both reconstructed and healthy ACLs. The inclusion of first echo (TE=0.1 ms) may led to errors in mono-exponential T2* estimation. A comparison of results between using all 4 TEs vs. using only 3 TEs (excluding first TE) would be meaningful.

5. As indicated in literature, there are significant changes during the first 6 months post-op. Some discussion/clarification are needed.

Reviewer #2: In the present prospective study, the authors Rikuto Y et al. use a UTE T2* mapping technique on a defined patient collective (10 female patients, 18 +/- 4 years of age) to assess the UTE T2* value changes within the tendon graft in the first postoperative year after ACL reconstruction with regards to the histological process of ligamentisation. The progressing ligamentisation has been shown to correlate with an increasing stability of the ACL reconstruction (although mainly in animal trials), which is an important factor with regards to return to physical activity as well as to avoiding the risk of early graft re-tear.

In correlation with other recent studies by Chu CR et al (Orthop J Sports Med. 2019) and Warth RJ and al. (Am J Sports Med. 2020), the presented UTE T2* values of the intraarticular segment of the ACL grafts drop significantly between 6 to 9 months postoperatively, without another significant change in the last quarter of the first postoperative year. The study also assesses the UTE T2* values of the femoral and tibial intraosseous segments of the ACL grafts, which are significantly lower than the intraarticular UTE T2* values, especially 6 months postoperatively. Compared to previously assessed UTE T2* values in uninjured ACL (see Okuda M et al., Acta Radiol. 2021), the intraosseous graft segments present similar values. In addition, a significant difference of UTE T2* values between the femoral and the tibial intraosseous segments of the ACL grafts is noted.

The authors conclude that the ligamentisation processes in the intraarticular and intraosseous segments of the ACL reconstructions differ significantly within the first 6 months postoperatively with a slowly increasing convergence after 9 to 12 months, which points towards a significant tendency for faster maturation intraosseously. Moreover, a significant difference is described in ligamentisation of the intraosseous segments in the first postoperative year, which is discussed to possibly be in relation to tunnel motion.

The authors present their findings and conclusions, as well as the study’s limitations, in a clear and concise manner. The correlating histological and imaging findings by other studies, added in the introduction and the discussion, give weight to the study’s overall conclusion.

The patient collective is explained to have been selected prospectively and includes patients operated on by one orthopaedic surgeon with a precisely explained surgical method. While the patient collective is small, consisting of just one gender and limited to a certain age, it is within the limit of the predetermined, calculated sample size.

The technique used for imaging acquisition, including its limitations (e.g. metal artefacts, magic angle effect), is explained in detail and is easily understandable, both in the introduction and in the method section. The measurement method as well as the statistical evaluation are well described and not too technical. As a radiologist, I especially appreciate the precise explanations with regards to the differences in MRI imaging and MRI imaging acquisition concerning this kind of tissue material.

In the last few years, mapping techniques in MR imaging have gained importance in the diagnostic assessment of tissue characteristics, and in particular of tissue pathologies (e.g. T1/T2 mapping of myocardium in cardiac imaging). UTE sequences have been increasingly applied in the diagnostics of the musculoskeletal system, given that they can reveal alterations in tissues with short to ultra-short transverse relaxation times, such as bone, ligaments or tendons. Tissues with moderate to long transverse relaxation times, e.g. cartilage, can already be reliably assessed with conventional MRI sequences (see also Chang E. et al, Journal of Magnetic Resonance Imaging 41:870–883 (2015)).

From a radiologist’s point of view, studies like the present authored by Rikuto Y et al. may give rise to more detailed studies, which will hopefully contribute to establishing these type of measuring methods in the daily routine of MRI imaging. The aforementioned is subject to the condition that the correlation between measured values and biomechanical changes prove to be significant to the clinical outcome of patients.

No major issues can be observed in the manuscript.

However, following minor issues could be considered:

- Page 12, Line 193: The measurements were taken by one orthopaedic surgeon as defined in the methodology. I found no mention of other evaluators in the text. It is not completely clear to me how the interobserver reliability could be assessed precisely in the results.

- Page 10 Line 156-160, Page 13/14 Line 207-211, Page 18 Line 281-284: The same, detailed passages of measurement and result explanations, which are included already in the adjacent text, can be found underneath the figures’ titles. They read repetitive, especially since the figures aren’t shown within the text in the downloaded PDF version. In my opinion, captions of figures should be kept succinct.

- Page 16 Line 248-252 / Page 17 Line 267-270: Another exact text passage repetition. While both passages fit within the context of the discussion, their proximity within the text impacts the fluency of the narrative. Paraphrasing of one passage may be better.

6. PLOS authors have the option to publish the peer review history of their article (what does this mean?). If published, this will include your full peer review and any attached files.

Reviewer #1: **Yes: **Xiang He

Reviewer #2: **Yes: **Dr. Maria Elena Misu

---

## [Author Response · Author response to Decision Letter 0]

4 Jun 2022

Abstract

Conclusion

- Is a bit of repetition according to the abstracts result section? Please provide any potential clinical benefit of your study or describe why this finding could be important for future research (see comment on discussion below).

- Conclusion in the abstract and the manuscript should be same.

Response: Thank you for pointing this out. We have revised the conclusions to highlight the potential clinical benefits of the study in the Abstract (Page 3, Lines 39) as well as main text (Page 21, Lines 377). We have also ensured consistency between the conclusions in the Abstract and those in the main text.

Material & Methods

-Please provide more information about the female patients included e.g. BMI, Height, side of injury, time from injury to surgery, maybe injury mechanism etc..

Response: Thank you for the recommendations. We have added the required information in the Patients and Methods section (Page 8, Lines 119).

Discussion

-The first paragraph of the discussion should contain the most important findings of your study. I would suggest using the actual conclusion as first paragraph for the discussion. Then rewriting the conclusion with reference to the clinical importance or future research topics you would suggest.

Response: Thank you for pointing this out. We have now revised the first paragraph of the Discussion section (Page 15, Lines 226) and the Conclusions section (Page 21, Lines 343) in accordance with your suggestions.

-I would like to make the manuscript more interesting to read for orthopedic surgeons having to deal with the RTA/RTS issues in daily routine.

-I would like to ask you to move figure 3 to the results section. Further, please explain in the methods section how “normal ACL” values were aquired.

Response: Thank you for your suggestions. We have moved Figure 3 to the Results section (Page 12, Lines 184) and added a reference citation pertaining to the measurement of normal ACL values in the Methods section (Page 10, Lines 148).

Limitations

-You did not provide data about the stability of knee after surgery. Orthopedic Surgeons well know that technical errors during surgery may result in decreased stability and that might affect the biological graft transformation. If you could provide data about the knee stability (clinical like Lachmann or KT 1000) that would improve the findings. If you don not have that data please mention this fact in the limitations.

Response: Thank you for the valuable insights. We are currently investigating the association of UTE-T2* values with clinical and functional outcomes (IKDC score and KOOS at 1 year postoperatively) of patients, including the subjects of the present study. Data regarding knee stability after surgery were insufficient in the present study. In the limitations section of the current manuscript, we have mentioned that investigation of the relationship of UTE-T2* values with clinical and functional outcomes is also important to provide ideas on safe return to sports in terms of maturation of reconstructed ACLs (Page 20, Lines 327).

-Please add information how the choice of enrolled participants could have influenced the results (only females..time between injury and surgery etc..)

Response: Thank you for pointing this out. We agree that selection bias is a concern in this study. We have mentioned this in the limitations paragraph and highlighted that future studies involving larger sample sizes and incorporating covariates that may affect the maturity of reconstructed ACLs are warranted (Page 20, Lines 328).

-Did metal remnants from tunnel drilling interfere with the scan /measures?

Response: Thank you for the pertinent question. If metal remnants were present, they would have a significant impact on the measurement as large artifacts. There were no such cases in the present study.

Reviewer #1: In this manuscript, authors investigated the maturation of reconstructed ACL using UTE-T2* imaging. This is a well-written manuscript. I do have some minor concerns:

1. Statistical Analysis: Only pairwise comparison of T2* values was conducted at three time points. As demonstrated in Figure 2, there are some clear trends of T2* over time, especially in intraarticular region. Correlation T2* values with recovery time would be interesting.

2. In Discussion, authors claimed that this is the first study to investigate regional graft maturation with UTE. There is at least one recent publication in this topic: Fukuda T, et al, Abbreviated quantitative UTE imaging in anterior cruciate ligament reconstruction, BMC Musculoskeletal Disords. 2019;20(1):426;

Response: Thank you for pointing that out. As you pointed out, previous studies have measured the T2* values of reconstructed ACLs separately in the joint and in the bony foramen. We have deleted the word “first” and rephrased the sentence in the revised manuscript (Page 15, Lines 226 and Page21, Lines 337).

3. In Discussion, authors claimed the long acquisition duration needed for bi-exponential UTE study. It would be true if using fully-sampled acquisition. As described in Fukuda et al paper, it is possible to reduce the time by partial acquisition.

4. The confounds of mono-exponential instead of bi-exponential should be discussed. The fraction of bound water is quite high in both reconstructed and healthy ACLs. The inclusion of first echo (TE=0.1 ms) may led to errors in mono-exponential T2* estimation. A comparison of results between using all 4 TEs vs. using only 3 TEs (excluding first TE) would be meaningful.

Response: Thank you for your valuable comments and suggestions. We have mentioned in the limitations that biexponential UTE-T2* mapping was deemed inappropriate for routine clinical MRI in this study because of its long acquisition time, although partial acquisition could have reduced the time. We have also discussed the confounds of using mono exponential mapping instead of biexponential mapping in the limitations (Page 20, Lines 321).

5. As indicated in literature, there are significant changes during the first 6 months post-op. Some discussion/clarification are needed.

Response: Thank you for the helpful comments. We have discussed the main histological findings pertaining to the changes during the first 6 months after surgery in the Discussion section (Page 16, Lines 249).

Reviewer #2

From a radiologist’s point of view, studies like the present authored by Rikuto Y et al. may give rise to more detailed studies, which will hopefully contribute to establishing these type of measuring methods in the daily routine of MRI imaging. The aforementioned is subject to the condition that the correlation between measured values and biomechanical changes prove to be significant to the clinical outcome of patients.

Response: Thank you for your detailed review and positive feedback for our study. We are currently investigating the association of UTE-T2* values with clinical and functional outcomes (IKDC score and KOOS at 1 year postoperatively) of patients, including the subjects of the present study. We will analyze and present the results once a sufficient sample is collected.

- Page 12, Line 193: The measurements were taken by one orthopaedic surgeon as defined in the methodology. I found no mention of other evaluators in the text. It is not completely clear to me how the interobserver reliability could be assessed precisely in the results.

Response: We apologize for the lack of clarity regarding interobserver reliability. We have now mentioned in the Patients and Methods section that another orthopedic surgeon (YY, Observer 2) independently performed measurements using the same method in order to assess interobserver reliability (Page 10, Lines 155).

- Page 10 Line 156-160, Page 13/14 Line 207-211, Page 18 Line 281-284: The same, detailed passages of measurement and result explanations, which are included already in the adjacent text, can be found underneath the figures’ titles. They read repetitive, especially since the figures aren’t shown within the text in the downloaded PDF version. In my opinion, captions of figures should be kept succinct.

Response: Thank you for pointing this out. We have revised each caption to be more concise.

- Page 16 Line 248-252 / Page 17 Line 267-270: Another exact text passage repetition. While both passages fit within the context of the discussion, their proximity within the text impacts the fluency of the narrative. Paraphrasing of one passage may be better.

Response: Thank you for pointing this out. We have now deleted and rearranged some text in the repetitive paragraph (Page 17-18, Lines 266-289).

---

## [Decision Letter · Decision Letter 1]

11 Jul 2022

Ligamentization of the reconstructed ACL differs between the intraarticular and intraosseous regions: A quantitative assessment using UTE-T2* mapping

PONE-D-22-05680R1

Dear Dr. Nakase,

We’re pleased to inform you that your manuscript has been judged scientifically suitable for publication and will be formally accepted for publication once it meets all outstanding technical requirements.

Kind regards,

Markus Geßlein, Ph.D.

Section Editor

PLOS ONE

Additional Editor Comments (optional):

Dear Authors,

thank you very much for submitting your fine work to PLOS ONE.

The manuscript is now accepted for publication. I would kindly ask you to further adress some minor changes requested by one of the reviewers boefore we proceed:

In this revised manuscript, authors have adequately addressed all my concerns. There are only some very minor changes:

1. Page 6, Line 78-81: Change into: " ... the conventional MRI signal intensity is affected by the image sequence and scanner hardware. Moreover, Tendon and ligaments normally have short T2 relaxation times leading to low MRI signal in conventional MRI protocols".

2. Page 6, Line 83: Relaxation time T2* reflects the intrinsic property of tissue and should be independent on image sequence and acquisition parameters.

3. Page 6, Line 84-85: Change the original sentence into: "These variables reflects the T2* relaxation of bounded water with collagen of tendons and ligaments, and an ideal to capture the changes in tissue structure and organization during ligamentization of reconstructed ACL".

4. Page 6, Line 88-89: Change the original sentence into: "...., which cannot be assessed in conventional gradient-echo based MRI assessment due to sub-ms T2* values of collagen-bound water, ....".

5. Page 20, Line 326: Adding one sentence: "Using mono-exponential T2* fitting, the estimated apparent T2* values as reported in this study may include the effect from changes of bound/unbound tissue water fraction during ACL ligamentization."

Thank you very much and congratulatuions!

Markus Geßlein, PhD

Reviewers' comments:

Reviewer's Responses to Questions

**Comments to the Author**

1. If the authors have adequately addressed your comments raised in a previous round of review and you feel that this manuscript is now acceptable for publication, you may indicate that here to bypass the “Comments to the Author” section, enter your conflict of interest statement in the “Confidential to Editor” section, and submit your "Accept" recommendation.

Reviewer #1: All comments have been addressed

Reviewer #2: All comments have been addressed

2. Is the manuscript technically sound, and do the data support the conclusions?

Reviewer #1: Yes

Reviewer #2: Yes

3. Has the statistical analysis been performed appropriately and rigorously? 

Reviewer #1: Yes

Reviewer #2: Yes

4. Have the authors made all data underlying the findings in their manuscript fully available?

Reviewer #1: No

Reviewer #2: Yes

5. Is the manuscript presented in an intelligible fashion and written in standard English?

Reviewer #1: Yes

Reviewer #2: Yes

6. Review Comments to the Author

Reviewer #1: In this revised manuscript, authors have adequately addressed all my concerns. There are only some very minor changes:

1. Page 6, Line 78-81: Change into: " ... the conventional MRI signal intensity is affected by the image sequence and scanner hardware. Moreover, Tendon and ligaments normally have short T2 relaxation times leading to low MRI signal in conventional MRI protocols".

2. Page 6, Line 83: Relaxation time T2* reflects the intrinsic property of tissue and should be independent on image sequence and acquisition parameters.

3. Page 6, Line 84-85: Change the original sentence into: "These variables reflects the T2* relaxation of bounded water with collagen of tendons and ligaments, and an ideal to capture the changes in tissue structure and organization during ligamentization of reconstructed ACL".

4. Page 6, Line 88-89: Change the original sentence into: "...., which cannot be assessed in conventional gradient-echo based MRI assessment due to sub-ms T2* values of collagen-bound water, ....".

5. Page 20, Line 326: Adding one sentence: "Using mono-exponential T2* fitting, the estimated apparent T2* values as reported in this study may include the effect from changes of bound/unbound tissue water fraction during ACL ligamentization."

Reviewer #2: The reviewed article has addressed all remarks made by both the editor and the reviewers, and the text reads fluently and concisely after the revision.

The patient collective is explained much better now, as is the trauma mechanism leading to the ACL repair in the first place.

The revised article also contains additional information on the histological changes that occur, and the article correlates them well to the signal changes and T2* values acquired in this study. The limitations to the study are also far more precise, and pointing towards the need of future studies with a broader patient collective.

One question that remains is whether the very small patient collective (mainly one gender and one age-range) as well as the choosing of only non-contact trauma mechanisms could potentially limit the concluded results to just one particular group of patients (e.g. young adults), and therefore not make it fully applicable to the range of patients with ACL repairs, esp. when it comes to the ideal time for the return to sports.

No major issues can be observed in the revised manuscript.

7. PLOS authors have the option to publish the peer review history of their article (what does this mean?). If published, this will include your full peer review and any attached files.

Reviewer #1: No

Reviewer #2: **Yes: **Dr. Maria Elena Misu

---

## [Editor Report · Acceptance letter]

14 Jul 2022

PONE-D-22-05680R1 

Ligamentization of the reconstructed ACL differs between the intraarticular and intraosseous regions: A quantitative assessment using UTE-T2* mapping 

Dear Dr. Nakase:

I'm pleased to inform you that your manuscript has been deemed suitable for publication in PLOS ONE. Congratulations! Your manuscript is now with our production department. 

Kind regards, 

on behalf of

Dr. Markus Geßlein 

Section Editor

PLOS ONE